

# Local villagers' perceptions of wolves in Jiuzhaigou County, western China

Yu Xu[1,3], Biao Yang[2,3] and Liang Dou[2]

[1] School of Resources and Environmental Sciences, Pingdingshan University, Pingdingshan, China
[2] Key Laboratory of Bio-Resources and Eco-Environment of Ministry Education, School of Life Sciences, Sichuan University, Chengdu, China
[3] These authors contributed equally to this work.

## ABSTRACT

While there have been increasing numbers of reports of human-wolf conflict in China during recent years, little is known about the nature of this conflict. In this study, we used questionnaires and semi-structured interviews to investigate local villagers' perceptions of wolves in Jiuzhaigou County, western China. We sampled nine villages with more frequent reports of wolf depredation to the local government, but included three villages near alpine pastures in which reports of depredation were less frequent. We sampled 100 residents, a subset of the local population who were more likely to have had experience with wolves. During the preceding three years, most families of the respondents grazed livestock on alpine pastures, and most of them reported that their livestock suffered from depredation by wolves. The mean value of the reported annual livestock loss rates was considerably higher in villages that reported depredation more frequently than in those with less frequent reports of depredation. Most respondents in the more frequently depredated villages perceived an increase in wolf populations, whereas many in the less frequently depredated villages perceived a decrease in wolf populations in their areas. People's attitudes towards wolves did not differ significantly between these two village categories. The majority of the respondents were negative in their attitude to wolves, despite a prevalent Tibetan culture that favors the protection of wildlife. People's negative attitude was directly related to the number of livestock owned by their family. Those with a larger number of livestock were more likely to have a negative attitude towards wolves. Factors such as village category, ethnicity, age and education level did not influence people's attitudes to wolves. We suggest that improved guarding of livestock and provision of monetary support on human resources and infrastructure may mitigate human-wolf conflicts in this region.

Corresponding author
Yu Xu, xuyu608@163.com

## INTRODUCTION

Conflicts between wolves and humans are common worldwide, because of human fear of wolves and in particular, financial loss due specifically to injury to and loss of livestock that wolves cause (*Mech & Boitani, 2003*; *Treves & Karanth, 2003*; *Bisi et al., 2007*). As wolf populations expand or humans encroach on their habitats, economic losses to wolves

increase and conflicts have become increasingly likely, presenting unique challenges for the conservation and management of wolves (*Mech & Boitani, 2003*; *Naughton-Treves, Grossberg & Treves, 2003*; *Bisi et al., 2007*; *Lescureux & Linnell, 2013*).

To conserve wolves, many researchers have conducted studies aimed at understanding the conflicts. Managers need to know the perceptions and attitude of local people to wolves and their conservation programs. In particular, by determining which people are more negative towards wolves, managers are potentially able to find solutions to improving people's tolerance towards wolves (*Treves & Karanth, 2003*; *Suryawanshi et al., 2013*). It has been pointed out that people's attitude depends mainly on their residence, age, gender, education and income level, and former experience with wolves (for a review of 38 surveys see *Williams, Ericsson & Heberlein, 2002*; *Ericsson & Heberlein, 2003*; *Naughton-Treves, Grossberg & Treves, 2003*; *Røskaft et al., 2007*; *Skogen & Thrane, 2007*). Yet, it is hard to find a widely accepted management policy, because of regional variation and different factors affecting attitudes (*Bjerke, Reitan & Kellert, 1998*; *Bisi et al., 2007*). Given this, one has to learn more about the characteristics of human-wolf conflicts and human attitudes towards wolves, especially for regions where conflicts have been reported frequently but available knowledge is very scarce.

China has a large wolf population which is mainly distributed in areas with relatively few anthropogenic changes, in the Qinghai–Tibet Plateau, the Mongolia Plateau and the Northeast Plain. In 1998, the number of wolves in China was estimated as about 6,000 individuals (*Wang, 1998*; *Yang, 2008*). In this year, wolves were listed as a vulnerable species in the China Red List (*Wang, 1998*), and since then, all hunting has been banned for this legally protected animal. In 2003, China was estimated to have a population of as many as 12,500 individuals (*Mech & Boitani, 2003*). We are not aware of any more recent estimates, although the number is likely to have increased substantially in recent years. In recent years in China, there have been increasing reports of injury and loss wolves caused especially to livestock, resulting in increase in human-wolf conflicts (*Yang, 2008*; *Zhang et al., 2010*; *CNC, 2012*; *Li et al., 2013*; *ScienceNet, 2013*). However, the published literature on this topic is scarce. There have not been any national policies relating to wolves other than some with indirect implications such as auctions of licenses for hunting wild animals (*BBC News, 2006*) and eco-compensation to mitigate human-wildlife conflicts (*Xinhuanet, 2014*; *Yunnan.cn, 2014*).

In this study, we examined local villagers' perceptions of wolves in Jiuzhaigou County, where wolf depredation on livestock has been reported increasingly and the local government is considering management plans for wolves. We aimed to determine wolf population trends, since there have been no data available on the wolf populations in this area. Furthermore, we aimed to determine the level of livestock depredation caused by wolves, and then how people's attitude toward wolves was related to socioeconomic variables, specifically religious belief (e.g., *Liu et al., 2011*) and livestock ownership (e.g., *Tuğ, 2005*), which are poorly understood.

## METHODS

### Ethics statement

The study conformed to the Declaration of Helsinki, and the Ethics Committee of Pingdingshan University approved the research protocol (Ref: 2012003). Verbal informed consent was obtained from all the subjects prior to participation.

### Study area

We conducted the study in Jiuzhaigou County (N 32°53′–33°43′, E 103°27′–104°26′; Fig. 1), Aba Tibetan and Qiang Autonomous Prefecture of northwestern Sichuan Province, western China. The county lies at the northeastern edge of Qinghai–Tibet Plateau and is famous for its Jiuzhai Valley National Park and the traditional cultures of its inhabitants. The area is 5,290 km², with an elevation ranging from 1,000 m to 4,500 m. The climate is subtropical to temperate monsoon with a mean annual temperature of 12.7 °C. Total annual rainfall is 550 mm, with 80% of rainfall occurring between May and October. The county comprises 17 townships and 120 villages, inhabited by Tibetan, Qiang, Hui, Han and other ethnic groups. In 2011, the county's population was 66,246, with a minority population (ethnic groups other than Han) of 25,090.

Jiuzhaigou County has 3,570 km² of forested lands (covering about 67% of the total area), and is the second largest forest area in Sichuan Province. It is rich in alpine grasslands, especially in the northern part, with an area of about 1,200 km² (*Chen, 2011*). Livestock grazing occurs mainly in the northern region. Yaks are the most common livestock species grazed, but there are a few sheep and goats. Livestock are herded to alpine pastures except during extreme winter weather when they are herded in the cropland around the villages or are stall-fed inside the villages. Livestock of each village graze in exclusive pastures. Families take turns at herding the entire village's stock. Commonly, a couple of people herd the livestock, with the use of one or two shepherd dogs occasionally. The livestock are usually left to range freely on the daytime. At night, the herders bring them back to protective corrals, which are poorly built with low walls and no roof.

Wolves are one of the most important animal species in the local ecosystems. Their large natural prey species are ungulate animals including *Elaphodus cephalophus*, *Capreolus capreolus*, *Capricornis sumatraensis*, *Naemorhedus goral*, and *Pseudois nayaur*; smaller prey include *Marmota himalayana*, *Lepus oiostolus*, and *Ochotona thibetana*, and some Galliformes such as *Tetraogallus tibetanus*, *Tetraophasis obscurus*, *Perdix hodgsoniae*, *Ithaginis cruentus*, *Pucrasia macrolopha*, and *Chrysolophus pictus*. However, prey abundance is low (*SPAFS, 2004*; *SCUSLS, 2011*). In the area, livestock depredation by wolves has been reported frequently in recent years, whereas there are few reports on wolf attacks on humans. The local people reported that wolves usually traveled in groups, and attacked livestock during both the day and night.

### Fieldwork

We carried out the fieldwork in April and May 2012. Following advice from the local forestry bureau, we conducted an interview survey in the northern region where many

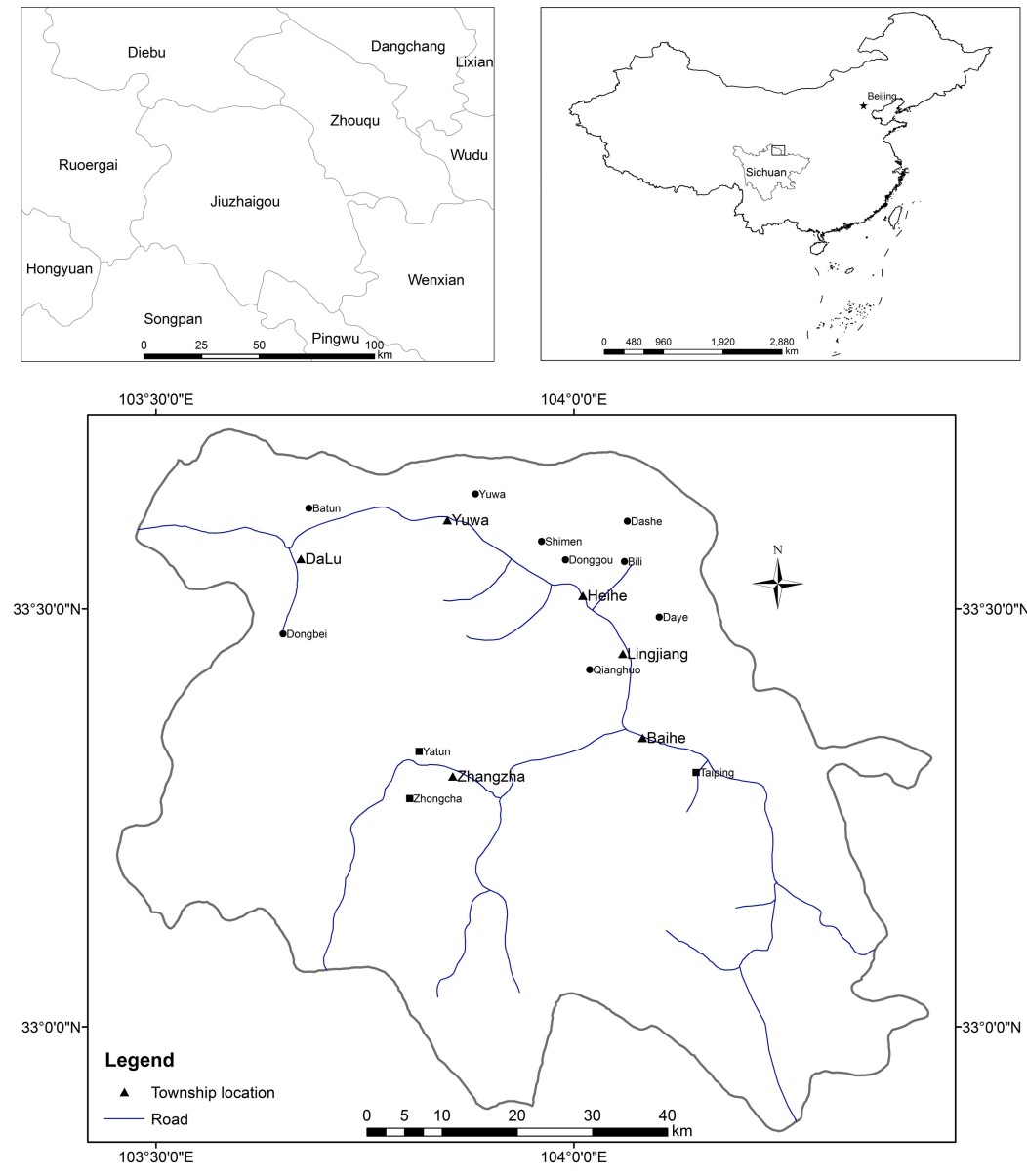

**Figure 1 Map showing the study area, Jiuzhaigou County, Sichuan Province, western China, as well as locations of villages investigated in the study.** The dark circles represent villages with more frequent reports of wolf depredation, and the dark squares villages with less frequent reports of wolf depredation

villages reported frequently livestock depredation by wolves to the local government. We sampled nine villages with frequent reports of depredation. For comparison, we also sampled three villages near alpine pastures in which reports of depredation were less frequent (Fig. 1). The people we interviewed were those who were 16 years or older and who grazed their livestock, collected herbs or mushrooms, or carried out other activities in forests and pastures, as they had a better knowledge of the population and activities of wolves (*Gros, 1998*). To foster a comfortable, non-intimidating information exchange

process with local residents, the interviews did not include any government staff, and we explained to local residents that the interview was for scientific research purposes only (*Kvale, 1996*).

During interviews, we used a semi-structured questionnaire to minimize the influence of the questions on the response (*Wengraf, 2001*). Interviews were performed orally with responses recorded immediately post-interview into the standardized questionnaire. First, we recorded respondents' personal characteristics, including ethnicity (a variable correlated with the religious belief system), gender, age and education level (three levels: "illiterate," "elementary school," and "secondary and above"). We did not ask their household incomes, because a pilot survey found it difficult to get the true value from respondents. Second, we asked for information about livestock in the past three years, including the annual number of livestock grazed by each family and the annual number of livestock depredated by wolves. Third, we asked their opinions about wolf population trends in their areas over the past 10 years ("increasing," "stable," and "decreasing"). Finally, we assessed their attitudes towards wolves. We included three questions as proxy measures for attitude: (1) "What do you think of wolves?"; (2) "What do you intend to do in response to livestock depredation by wolves?"; (3) "Do you think that wolves should be protected?" If a person thought wolves were bad and detrimental, wanted to kill wolves who were attacking their livestock, and did not wish to protect wolves, we considered that the person was negative towards wolves; if the opposite were the case, we considered the person to have a positive attitude towards wolves. If a person had no strong opinion to the questions, we considered the person to have a neutral attitude towards wolves. If the respondent showed a positive attitude in response to some questions and a negative attitude in response to others, we defined this as a mixed response.

## Data analysis

We first calculated descriptive statistics of responses to analyze the basic information from the interviews. We used to T-tests and Chi-squared tests of independence (Fisher's exact tests were employed if any expected frequency was below 1 or more than 20% of expected frequencies were less than 5) to compare differences of responses between two village categories, those with more frequent reports of wolf depredation and those less frequent reports of wolf depredation.

We then examined how people's attitudes towards wolves were affected by variables of personal characteristics (including ethnicity, age, and education level; gender was not included in the analysis because of only a few respondents were women), and by variables relating to livestock (i.e., annual number of livestock grazed and annual percentage of livestock depredated by wolves). We included all villages but used village category as an additional prediction factor. We excluded one sample with mixed opinion about our measures and pooled positive and neutral attitudes due to the small number of responses in these categories. We also excluded two samples with missing values. We conducted a binary logistic regression, where a totally negative attitude was scored as 1 while a positive or neutral attitude was scored as 0. In the regression, annual percentage

**Table 1** Comparison of respondents' personal characteristics between villages with more frequent reports of wolf depredation and those with less frequent reports of wolf depredation.

| Village category | Mean age (range) | Proportion of ethnic groups | | Proportion of education levels | | |
|---|---|---|---|---|---|---|
| | | Tibetan | Han | Illiterate | Elementary school | Secondary and above |
| Villages with more frequent reports of depredation ($n = 9$ villages, and 81 respondents) | 44 (16–81) | 49.4% (40/81) | 50.6% (41/81) | 34.6% (28/81) | 45.7% (37/81) | 19.8% (16/81) |
| Villages with less frequent reports of depredation ($n = 3$ villages, and 19 respondents) | 46 (23–77) | 52.6% (10/19) | 47.4% (9/19) | 42.1% (8/19) | 36.8% (7/19) | 21.1% (4/19) |
| Statistical tests | $t = -0.536$, df $= 98$, $P = 0.593$ | $\chi^2 = 0.065$, df $= 1$, $P = 0.799$ | | $\chi^2 = 0.529$, df $= 2$, $P = 0.768$ | | |

of livestock depredated was considered 0 if there was no livestock grazed (note that this applied only to the regression and not to the previous descriptive statistics). Categorical variables including village category, ethnicity and education level were converted into a set of dichotomous, dummy-coded variables. We set "villages with more frequent reports of wolf depredation" as the reference for village category, "Tibetan" as the reference for ethnicity, and "illiterate" as the reference for education level.

We used the Akaike information criterion corrected for small sample size (AICc) to compare statistical models constituting different combinations of variables. We calculated ΔAICc, which means the difference between the model with the lowest AICc and the other models in the model set. The model with the lowest AICc was considered as the best model, when ΔAICc between it and the next best model was larger than two (*Burnham & Anderson, 2002*). However, if there were some models whose ΔAICc was ≤2 (which means that these models had equivalent support to explain the data), we used a normal model averaging approach over all candidate models to get parameters and error estimates (*Burnham & Anderson, 2002*; *Anderson, 2008*). We calculated the 95% confidence interval (95% CI) of parameter estimate and the odds ratio (OR) of the effects for each variable. We also estimated the relative importance ($w_+$) of a given variable by summing the Akaike weights of all models containing the variable, and compared variables by examining the ratio of $w_+$. Variables were considered as associated statistically with the response variable, when their 95% CIs excluded the zone value (*Grueber et al., 2011*). All analyses were performed on R 3.0.0 (*R Development Core Team, 2013*).

## RESULTS

In total, we interviewed 100 residents, of which 81 belonged to villages that reported wolf depredation more frequently, and 19 to villages with less frequent reports of depredation. The respondents' personal characteristics did not differ significantly between the two village categories (Table 1). Overall, they averaged 44 years of age, with 36% illiterate,

**Table 2 Comparison of estimates of livestock ownership and livestock depredation between villages with more frequent reports of wolf depredation and those with less frequent reports of wolf depredation.**

| Village category | Percentage of families with livestock grazed | Average annual number of livestock owned per family (range) | Percentage of families with livestock depredated by wolves | The reported annual livestock loss rate to wolves (range) |
|---|---|---|---|---|
| Villages with more frequent reports of depredation ($n = 9$ villages, and 81 respondents[a]) | 86.3% (69/80) | 41 (4–200) | 82.6% (57/68) | 21.7% (0–70%) |
| Villages with less frequent reports of depredation ($n = 3$ villages, and 19 respondents) | 73.6% (14/19) | 53 (3–200) | 64.3% (9/14) | 11.7% (0–30%) |
| Statistical tests | Fisher's exact test $P = 0.184$ | $t = -0.981$, df = 81, $P = 0.330$ | Fisher's exact test $P = 0.134$ | $t = 1.180$, df = 80, $P = 0.074$ |

**Notes.**
[a] No data values were recorded for livestock ownership and livestock depredation in one sample, and there is one missing value for livestock depredation in another sample.

**Table 3 Comparison of respondents' opinions about wolf population trends and attitudes towards wolves between villages with more frequent reports of wolf depredation and those with less frequent reports of wolf depredation.**

| Village category | Proportion of opinions about wolf population trends | | | Proportion of attitudes towards wolves | | | |
|---|---|---|---|---|---|---|---|
| | Increasing | Decreasing | Stable | Negative | Positive | Neutral | Mixed |
| Villages with more frequent reports of depredation ($n = 9$ villages, and 81 respondents) | 79.0% (64/81) | 16.0% (13/81) | 4.9% (4/81) | 86.4% (70/81) | 7.4% (6/81) | 4.9% (4/81) | 1.2% (1/81) |
| Villages with less frequent reports of depredation ($n = 3$ villages, and 19 respondents) | 47.4% (9/19) | 52.6% (10/19) | 0 (0/19) | 73.7% (14/19) | 26.3% (5/19) | 0 (0/19) | 0 (0/19) |
| Statistical tests | Fisher's exact test $P = 0.005$ | | | Fisher's exact test $P = 0.116$ | | | |

44% with elementary education, and 20% with secondary or higher education. Half of the respondents were Tibetan, and the other half were Han.

During the preceding three years, most families of the respondents owned livestock that they grazed on alpine pastures. Most of them reported that their livestock suffered from depredation by wolves (Table 2). The two village categories we defined did not differ significantly in livestock ownership and percent of families experiencing depredation; however, there was a trend toward higher mean annual livestock loss rate in villages that reported depredation more frequently compared with those with less frequent reports of depredation (Table 2).

Most respondents in the more frequently depredated villages perceived an increase in wolf populations, whereas more than half of respondents in the less frequently depredated villages perceived a decrease in wolf populations in their areas (Table 3). With respect to people' attitudes towards wolves, there was no significant difference between the two village categories (Table 3). The majority of the respondents were negative in their attitude to wolves. All the people who were negative towards wolves mentioned that livestock loss

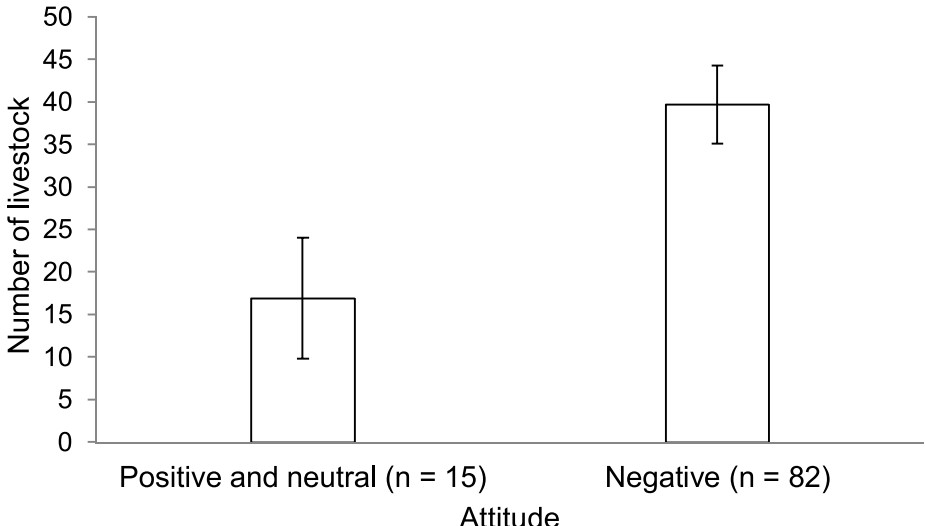

**Figure 2** Mean annual number of livestock ±1 SE grazed by families of respondents who had different attitudes towards wolves.

caused by wolves was the main reason for their attitude. Therefore, they wanted to kill wolves and did not wish to protect them. Only very few (3.6%, 3/84) people talked about fear of wolves, and no one mentioned an event of wolves attacking humans.

We constructed 64 candidate logistic regression models with six variables, where we excluded one sample with mixed opinion towards wolves and two samples with missing values, to predict variation of people' attitudes towards wolves. As there were 13 models with ΔAICcs ≤ 2, we used a model averaging approach to calculate estimates for variables. The results showed that the number of livestock owned was the most important predictor, with the other five variables having lower relative importance to it (Table 4). Only the number of livestock owned was statistically related to attitudes of the respondents towards wolves, as its 95% CI of the parameter estimate excluded the zero value. People with a more negative attitude towards wolves owned larger numbers of livestock, on average (Fig. 2). For each additional animal owned, people were, on average, 1.032 times more likely to have a negative attitude towards wolves (Table 4). Variation of attitudes towards wolves was not predicted by village category, the percentage of livestock depredated by wolves and factors associated with personal characteristics (i.e., ethnicity, age, and education level).

## DISCUSSION

Understanding the perceptions of local people living adjacent to wildlife habitats of wildlife-human interactions is important in the conservation of large carnivores, because they are apt to provide reliable information about wildlife (*Treves & Karanth, 2003*; *White et al., 2005*). However, it is difficult for the public to estimate wolf population sizes (*Bjerke, Reitan & Kellert, 1998*). In this study, we instead asked local people's opinions about wolf population trends. Similar studies have been previously conducted on other large carnivores, for example the Asiatic black bear *Ursus thibetanus* (*Liu et al., 2011*).

**Table 4  Model-averaged coefficients and relative importance calculated for variables explaining variation in attitude of respondents towards wolves.**

| Variable | Estimate | SE | Lower 95% CI | Upper 95% CI | OR | z value | wi+ |
|---|---|---|---|---|---|---|---|
| Intercept | 0.175 | 1.115 | −2.010 | 2.360 | 1.191 | 0.156 | |
| ANL | 0.032 | 0.016 | 0.001 | 0.062 | 1.032 | 2.018 | 0.93 |
| Ethnicity_Han[a] | 1.082 | 0.747 | −0.382 | 2.546 | 2.951 | 1.432 | 0.53 |
| Village category_less frequent reports of wolf depredation[b] | −0.945 | 0.687 | −2.293 | 0.402 | 0.389 | 1.358 | 0.46 |
| APL | 2.144 | 1.806 | −1.395 | 5.683 | 8.534 | 1.173 | 0.44 |
| Age | 0.025 | 0.024 | −0.022 | 0.072 | 1.025 | 1.038 | 0.38 |
| Education level_secondary and above[c] | 0.370 | 0.870 | −1.335 | 2.075 | 1.447 | 0.42 | 0.32 |
| Education level_elementary school | 0.599 | 0.674 | −0.721 | 1.920 | 1.821 | 0.878 | — |

**Notes.**

[a] "Tibetan" was the reference category.

[b] "Villages with more frequent reports of wolf depredation" was the reference category.

[c] "Illiterate" was the reference category.

ANL, annual number of livestock the respondent's family grazed; APL, annual percentage of livestock depredated by wolves; OR, the odds ratio.

Most of the people we interviewed reported an increase in wolf populations in their areas in the preceding 10 years. The apparent increase in abundance may reflect effective protection and population recovery of the wolf in the wild since the prohibition of guns in 1996. This may have resulted in the increased livestock depredation and human-wolf conflicts as reported. Increase in human-wolf conflict may also be a result of human encroachment on wolves' natural habitats (*Naughton-Treves, Grossberg & Treves, 2003*) and ongoing degradation or loss of habitat (*Yang, 2008*). In this case, wolves may more frequently encounter and prey on livestock as their natural prey populations have declined. Nevertheless, more than half of respondents in the less frequently depredated villages perceived a decrease in wolf abundance in their areas. Insights that we have gained in these villages indicate that some people illegally killed wolves in their areas. It should be noted that livestock losses were self-reported in the interviews, and the reported magnitude of losses may differ from reality. It was hard to verify the magnitude of these reported losses in the present study, and thus we suggest that additional research is needed on this topic.

As reported by some previous studies in other regions (e.g., *Ericsson & Heberlein, 2003*; *Naughton-Treves, Grossberg & Treves, 2003*; *Tuğ, 2005*; *Røskaft et al., 2007*), the local population had a negative attitude to wolves. Attitudes toward wolves are strongly driven by physical and behavioral characteristics of wolves as well as by some cultural and historical associations such as human fear of wolves (*Kleiven, Bjerke & Kaltenborn, 2004*; *Bisi et al., 2007*; *Suryawanshi et al., 2013*). In our study, the local people's explanation for negative attitude was livestock loss due to wolves, while very few indicated fear of wolves. Despite widespread fear of wolves, attitudes may differ between areas and groups as a result of different awareness of the biological characteristics of wolves, including avoidance of humans (*Bisi et al., 2007*; *Yang, 2008*).

However, our finding, that variation of attitudes was not explained by personal characteristics, is inconsistent with previous studies in which people's attitudes towards wolves

differed significantly according to their age, education level and income (e.g., *Ericsson & Heberlein, 2003*; *Naughton-Treves, Grossberg & Treves, 2003*; *Tuğ, 2005*; *Røskaft et al., 2007*). We found that Tibetan people were as negative as Han people, and this was not expected, as Tibetan groups, being Buddhist, find it easier to accept the protection of wildlife than Han people who have no dominant religion (*Eckel, 1998*). A similar result was also reported by *Liu et al. (2011)* in the study on human-bear conflicts of western Sichuan. It is possible that increase in conflicts between wolves and the local villagers at our study area might have resulted in negative public opinion. We suspect, though, that Tibetan people would be less likely to attack wolves in response to livestock loss or to engage in poaching, because of their belief that killing wildlife could negatively affect their resurrection in the afterlife (*Eckel, 1998*; *Liu et al., 2011*).

Little research has examined the potential links between attitude and variables concerning livestock. This study revealed that people with relatively large numbers of livestock were more likely to have a negative attitude towards wolves than those with smaller numbers of livestock. To our knowledge, there is no previous literature that has reported this phenomenon. In addition, we did not find that people who had lost a larger percentage of their livestock to wolf depredation showed a more negative attitude to wolves, as concluded by some previous studies concerning the wolf conflicts (*Williams, Ericsson & Heberlein, 2002*; *Ericsson & Heberlein, 2003*; *Naughton-Treves, Grossberg & Treves, 2003*; *Tuğ, 2005*). We acknowledge that assigning zero predation to informants who had no livestock would artificially reduce the predation intensity, and thus may have affected the relationship between livestock losses and attitude towards wolves. Unfortunately, because of small samples, we cannot further test the effect of percent loss using only those informants who had livestock.

As an explanation, we suggest that the current finding might be associated with the fact that the local people were impoverished and had seldom been compensated for their losses, while livestock mortality by wolves was a relatively common occurrence. It is expected that people who grazed a larger number of livestock and whose main source of income was from livestock, would be negative towards anything that may cause loss of their livestock and threaten their income. Even if wolves had not killed their livestock in the past, they would still have a negative attitude to wolves as they felt that no one could guarantee the safety of their livestock in the future. In contrast, people with a smaller number of livestock would expected to be neutral or positive in their attitude to wolves, as usually they were able to obtain income from other sources and the economic benefits from livestock accounted only for a small part of their incomes.

## Management implications

To mitigate future human-wolf conflicts, we must reduce livestock losses of local people who suffer from wolf depredation. In our study site and related areas, a large livestock group is herded commonly by a couple of people. Younger people are not willing to take up this lifestyle. Furthermore, existing corrals are poorly built. Ineffective guarding of livestock might have facilitated depredation by wolves (*Jackson, 2000*; *Treves & Karanth,*

*2003*; *Li et al., 2013*). Therefore, we suggest that the best approach at present should be to improve guarding of livestock in the context of local cultures and conditions, for example, increasing the number of herders, developing expertise in herding, and building wolf-proof corrals using local materials (see *Namgail, Fox & Bhatnagar, 2007*).

Eco-compensation in mitigating human-wildlife conflicts has been increasingly emphasized by the government in recent several years (*Xinhuanet, 2014*; *Yunnan.cn, 2014*). Public education on wolf conservation has been conducted in our study area, but there has been no any provision of monetary compensation for herders who lost livestock to wolves. The local forestry department mentioned many obstacles, such as the difficulty of verifying the magnitude of livestock losses reported by local villagers. As an alternative approach, we could invest these monies in human resources and infrastructure, such as training herders and improving corrals. This will distribute the benefit equitably (*Namgail, Fox & Bhatnagar, 2007*). Furthermore, initiation of a livestock insurance program guided by the government, a measure that has proved effective in the India's Trans-Himalayan region (*Mishra et al., 2003*), is encouraged for a long-term management.

There are other alternatives such as relocation or limited removal of problem wolves (e.g., *Mech & Boitani, 2003*; *Treves & Karanth, 2003*; *Bradley et al., 2005*), and change of local livelihood (e.g., *Jackson, 2000*; *Conforti & de Azevedo, 2003*; *Li et al., 2013*). The local government is considering employing armed police to kill problem wolves. Although a reported increase in the wolf populations and in livestock depredation by wolves in our study area, causal relationships between them are not clear. In addition, there is no scientific information on wolf population sizes. Therefore, this measure will require further data on wolf population sizes and their relations with livestock depredation. The local government is also assisting herders to attempt to increase incomes from alternative sources, for example eco-tourism and the cultivation of economically important alpine plants, which aim to reduce their dependency on livestock. Two of the 12 villages we interviewed seemed to have been moving toward a more positive attitude towards wolves. However, it should be noted that local people might resist directions from authorities. A shift to other areas may also have different environmental impacts. The forms of income generation should be implemented and sustained selectively through existing institutions (*Jackson, 2000*).

## CONCLUSIONS

To conclude, this study investigated local villagers' perceptions of wolves in Jiuzhaigou County, western China. Most people, especially in villages with more frequent reports of wolf depredation to the local government, reported an increase of wolf population and thus increased livestock losses to wolves. People were generally negative towards wolves, despite a prevalent Tibetan culture that favors the protection of wildlife. These with a larger number of livestock were more likely to have a negative attitude towards wolves. In term of conservation management, we suggest that improved guarding of livestock and provision of monetary support on human resources and infrastructure may mitigate human-wolf conflicts in this region. Our study provides insights into management of human-wolf conflicts in western China.

## ACKNOWLEDGEMENTS

We would like to thank Jiuzhaigou Forestry Bureau for help and support on this research. We thank Xiaodong Gu (Sichuan Wildlife Resource Survey & Conservation Management Station) for helpful discussion on designing this survey, and Shulian Yang (Baihe Provincial Natural Reserve), Li Deng (Jiuzhaigou National Natural Reserve) and Yanmei Li (Gonggangling Provincial Natural Reserve) for assistance with collecting data. Thanks also to Man Zhang (Sichuan University) and Hanqiu Yue (Pingdingshan University) for their work on GIS-based mapping, and Pingjia Que (Beijing Normal University) for guideline with statistical analysis. Thanks especially to Donald L. Kramer and three anonymous reviewers who provided constructive suggestions to improve the manuscript.

### Funding

This study was partially supported by the National Natural Science Foundation of China (No. 31301896), and the Scientific Research Foundation for High-level Talents of Pingdingshan University (No. 2012006). The funders had no role in study design, data collection and analysis, decision to publish, or preparation of the manuscript.

### Grant Disclosures

The following grant information was disclosed by the authors:
National Natural Science Foundation of China: 31301896.
Scientific Research Foundation for High-level Talents of Pingdingshan University: 2012006.

### Competing Interests

The authors declare there are no competing interests.

### Author Contributions

- Yu Xu conceived and designed the experiments, performed the experiments, analyzed the data, wrote the paper, prepared figures and/or tables, reviewed drafts of the paper.
- Biao Yang conceived and designed the experiments, performed the experiments, analyzed the data, contributed reagents/materials/analysis tools, wrote the paper, prepared figures and/or tables, reviewed drafts of the paper.
- Liang Dou performed the experiments, reviewed drafts of the paper.

### Human Ethics

The following information was supplied relating to ethical approvals (i.e., approving body and any reference numbers):

The Ethics Committee of Pingdingshan University approved the research protocol (Ref: 2012003), according to the principles expressed in the Declaration of Helsinki. The Ethics Committee also agreed that all the subjects give a verbal informed consent prior to participation.

## Supplemental Information

Supplemental information for this article can be found online at http://dx.doi.org/10.7717/peerj.982#supplemental-information.

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
