# Peer review of "Local villagers’ perceptions of wolves in Jiuzhaigou County, western China"

_PeerJ, doi:10.7717/peerj.982_

## Round 0.1 · original submission · Minor Revisions

Editor’s Comments

Both reviewers indicate that your manuscript contributes to our limited knowledge concerning human attitudes toward wolves in Asia, although both suggest that the contribution could have been greater if you had used alternative approaches. Both reviewers have suggestions for improving the analysis and presentation, and I have some additional suggestions of my own. I am therefore encouraging you to make appropriate revisions to address the concerns of the reviewers and submit a revised version to PeerJ.

Reviewer 1 indicates that you should have addressed human fear of wolves or at least state why you did not address this topic.

This reviewer would like a more explicit statement about whether there is any independent information about wolf population size and conservation status and about the injury caused to humans and livestock in the study area and broader region. If there is no such information, you should state this explicitly.

This reviewer indicates that semi-structured interviews might have provided additional information to provide a more complete overview of people’s attitude toward wolves. I realize that you cannot change your sampling method, but it is possible that you can supplement the information provided by insights gained in informal discussions with the respondents.

The reviewer also suggests that you should frame your suggestions regarding mitigation in the context of how limited your understanding of the origins of the conflict remains. Perhaps adding more indication of additional information that could guide conflict reduction would be helpful.

Reviewer 2 suggests that you should be careful to explain that livestock losses are self-reported in the interviews. Is there any way in which you can verify the magnitude of these reported losses?

This reviewer also suggests that you attempt to use more precise and quantitative terms rather than general descriptive terms whenever possible.

Furthermore, this reviewer feels that you have not adequately integrated the large available literature on this topic into your findings. I note that for some contributions of your study you have related your findings to the literature (the relationship between magnitude of livestock losses and negative attitude, the lack of a relationship between personal characteristics and negative attitude, potential mitigation strategies). For other topics, however, you have not indicated whether there is any comparable literature and how it affects your interpretation. These topics include public perception of changes in wolf abundance and the relationship between number of livestock owned and a negative attitude toward wolves. For the topics where you have not incorporated literature, please review available literature to integrate previous findings or indicate that previous researchers do not appear to have examined that topic. For the topics where you have incorporated some literature already, please consider whether there are other important references and whether you have adequately related your findings to previous work. For contributions where there is no previous literature as far as you know, state this explicitly as it emphasizes the originality of your contribution.

This reviewer questions your use of logistic regression because of concerns regarding the categorization of responses, the high proportion of negative attitudes, and recently increased awareness of the risk of incorrect conclusions from a probabilistic analysis. The reviewer proposes the use of content analysis or another social science method. Although I do not disagree with the reviewer’s concerns, you have used an established approach and I would not insist that you change your entire analysis for this manuscript. However, you should clarify how easy or difficult it was to categorize responses and, for each finding, critically address the strength of the conclusion in your Discussion. I agree that a more complete Table 1 would help the reader understand your analysis and findings. You could look at some current publications using a similar analysis and see what information is normally presented in the table.

Reviewer 2 also suggests clarification of the study population and sampling procedure. In the Abstract, it is important to indicate that you sampled a subset of the local population who were more likely to have had experience with wolves. Selection of this sub-group does not sound like stratified random sampling, but you might have been referring to the distribution of sampling among villages or some other aspect. Please clarify your sampling procedure.

This reviewer also requests a more in-depth discussion, including consideration of alternative explanations for your findings and consideration of broader implications of your proposed mitigation strategies.

Additional comments from the editor:

When introducing the study area, it would be useful to indicate which species of livestock are kept, the husbandry practices (for example are the animals left to range freely or are they fenced, are they guarded by people or dogs, are they returned to a protective corral or barn at night). What are the natural prey species of wolves in the area? Indicate whether the wolves primarily diurnal or nocturnal in this area if it affects the risk of livestock mortality?

When interpreting the effects of number of livestock owned and proportion killed, it does not seem evident that loss to wolves would be less important for those with fewer animals. Isn’t it possible that those with fewer animals might be poorer so that the loss would be more important? Also, is there any relationship between the number of livestock and proportion lost? For example, a large herd might be better protected or more able to resist a wolf attack. A relationship between herd size and proportion killed might help interpret the observation that the number of livestock but not proportion lost is related to attitude toward wolves.

In the Discussion, please try to take a more self-critical approach to your findings and indicate the strength of the pattern and potential confounding effects.

I think that assigning 0 predation to informants who had no livestock will artificially reduce the predation intensity. In your methods, be sure that it is clear this applied only to the regression and not to the descriptive statistics. (It does become clear in the Results, but you should not raise concerns in the reader’s mind in the Methods.) It is possible that assigning 0 predation to respondents that had no livestock may have affected the apparent relationship between depredation losses and attitude toward wolves. Is there any way you can test the effect of percent loss using only those informants who had livestock, perhaps by carrying out a supplementary analysis?

In the Results, give ranges as well as means for the number of livestock owned and the percent depredation because these are independent variables in your regression and the range of values could have an important effect.

L132. Fig. 2 shows that people with a more negative attitude owned more livestock than people with a neutral or positive attitude, not the other way around as you stated. This is also incorrectly restated in the Discussion L148. If you are basing this statement on your regression analysis rather than Fig. 2, you need to rephrase it something like, ‘People with larger numbers of livestock were more likely to have a negative attitude toward wolves’.

L145. It is not clear what you mean by wolves ‘relocating to private lands’. Where were they previously?

On the headings of the supplementary data table, Annual is not spelled correctly.

In general, the quality of the English is quite good. However, there are some errors in grammar and spelling that should be corrected. I will email you a pdf of your manuscript with yellow highlighting to indicate problem areas because there are too many to list here. In many cases, I added a comment box to indicate alternative wording or punctuation. If you are not certain what the problem is, please contact me by email.

Reviewer 1 ·

Basic reporting

In the Introduction, when defining the conflict between wolves and humans, authors neglect strong emotions toward wolves. It is known from earlier studies that the attitude formation is strongly driven by fear of wolves as well as the specific characteristics of the species. Fear of wolves is a common phenomenon in the whole distribution range of the species and it needs to be acknowledged also here. Unfortunately, it seems that this lack in defining the conflict has restricted authors in the material and method section too. If this has been intentionally done by the authors and is, as such, a result of choice, then this choice should be clearly clarified and an explanation for this must be given.

In spite of this criticism, the manuscript includes results that are relevant in terms of the objectives set in the Introduction; to determine the wolf population trends and the level of livestock depredation caused by wolves and to examine peoples’ attitudes towards wolves in terms of socioeconomic variables, religious beliefs and ownership of livestock.

Authors need to indicate the estimates on wolf population in China as well as in the study area of the Jiuzhaigou County in the Introduction. If there are no reliable estimates available, this should be clearly stated in the text. Also, the conservation status of wolves in China needs to be indicated.

Experimental design

This research and its findings are a good example of a basic study. In particular the topic of attitudes toward wolves at local level is very important, and research in different countries and in different contexts is needed. In gathering material semi-structured interviews offer an excellent opportunity for the researchers to output a thick description of attitudes in its context. However, this is not the case in this research. Instead, authors have had very basic approach with questions limited to three in assessing the attitudes. Method is limited to quantitative measures whereas more holistic qualitative approach is missing. Valid information is missing for understanding the conflict and attitudes.

However, keeping in mind the purpose of the manuscript, relevant results are gained with selected methods.

Validity of the findings

Main concern is about the sections of Discussion and Conclusions in the context of conflict resolution. As the authors are Chinese they also have a basic knowledge on national and local preconditions of planning the wolf policies. These preconditions should be somehow included in the text. By being unfamiliar with the carnivore conflict circumstances in China I would be interested in the relation and significance of the economic losses that wolves cause. Are there any known wolf attacks on humans and are those cases well documented? Human injuries caused by wolves can pose a several threat to human welfare, and therefore, they cause large scale economic losses. In the Conclusions authors do not try to ask the question, if is it possible to harvest wolves or intimidate them, and therefore get wolves to be afraid of humans.

Authors suggest that local peoples should be adjusted to reduce their dependency on livestock in the long-term. How about local defiance toward authorities, who are going to implement these actions? If fear is one of the main motives to cause negative attitudes toward wolves, will local people change their attitudes towards wolves regardless of their livelihood? This is why it is short-sighted to draw this kind of conclusion because no holistic approach on attitudes toward wolves has been conducted in this study. These limitations should be stated in the Discussion.

Additional comments

No comments

Reviewer 2 ·

Basic reporting

Scientific validity - questionable (see below)
Suitability to join the scholarly literature - This manuscript requires fairly major editing, a more intensive literature review, and the analysis is overly simplistic. The manuscript contains multiple terms that probably shouldn't be used or, at least, need to be defined (e.g., severe, fairly common, good ways, large, reasonable, hopefully). Authors need to report findings as self-reported livestock losses (depredations) because reported losses may differ dramatically from reality (e.g., L123; many more examples exist, such as the Abstract). Author's do not adequately incorporate the large volume of literature on this topic from all around the world. Finally, I have concerns regarding experimental design (see Experimental Design comments).

Experimental design

The data collection approach (semi-structured interviews) does not fit well with the data analysis approach (logistic regression), which requires condensing highly complex data into qualitative categories that can be subject to interpretation. The data analysis approach is problematic because of the fact that most respondents were negative (84) and only a few were positive or neutral (15). This severely limits the statistical inferences of the study. Furthermore, many researchers have challenged the validity of step-wise model building using significance as it can lead to spurious results.

It would have been more appropriate to use content analysis and explore the intricacies of specific responses to the three main questions, of another qualitative social science method. Furthermore, many of the variables were not adequately assessed - assigning default values to respondents who did not have a particular response associated with a given variable is not appropriate - particularly given the variety (usually two possible outcomes) of potential responses.

Education level of the respondents was reported as achieving only 80% of respondents - what were the rest?

The authors should make note that their survey was biased towards those more likely to have experiences, particularly, negative experiences with wolves as they selected for individuals who lived near the forest and grazed livestock or did other activities near the forest - thus, these results represent a particular unique subset of the population and are definitely not representative and need to be reported as such (this is why such a large percent of individuals owned livestock). Furthermore, this is not stratified random. This is a biased interview approach. Which is ok, but needs to be reported as such, since it represents a particular unique subset of the population.

Validity of the findings

No Comments

Additional comments

This paper is interesting as little has been reported on wolf-human interactions and attitudes regarding wolves in Asia. However, besides that it is very simplistic and analyses are questionable.

Some of the alternative hypotheses proposed by the authors to explain their findings have some obvious additional alternative hypotheses (e.g., perception of increasing wolf numbers could be due also to a reduction in wild prey, changing livelihoods of locals, changing cultural perspectives of locals, etc). Lastly, reducing dependence on livestock only shifts livestock production to different areas - which may have different environmental impacts. The authors should further discuss the potential role of compensation and other alternatives such as, education and marketing to increase tolerance or limited removal of problem wolves by governmental employees or locals.
Table 1: You had so few variables why not show the output for all models considered so we can better understand what you did and how you are interpreting your data.
Figure 1: Provide some geographical information to help people unfamiliar with China understand where you are located (particularly in the inset)
Figure 2: What are the error bars?

---

## Round 0.2 · Minor Revisions

Overall, the revisions to the manuscript have addressed all the major issues raised by previous reviewers and the editor.

However, because the analysis was changed, it was necessary to find a reviewer able to assess the new information theoretic approach. This reviewer was asked to focus only on this aspect of the statistics. The reviewer confirmed the validity of the overall approach, but suggested that some clarifications and justifications were needed. I would like to elaborate on one point made by this reviewer. He or she noted the unusual procedure of using the 90% CI instead of 95% and requested a justification. Looking at the data in Table 1, I suspect that 90% was used because the effect of number of livestock would not be statistically significant if 95% was used. If there is no independent, a priori justification for using 90%, I suggest that the authors return to 95% CI and in their results indicate that ANL approached significance and was the most important variable.

Furthermore, with the revised description of village selection, I have a concern about potential bias in the study design. I am not sure I understand the procedure for village selection on L93-95. As written, the text implies that you selected villages that were reported to have high rates of wolf predation on livestock but included some villages that had low reported predation rates. If this is the case, there is a problem with the study design because the proportion of high and low predation villages you selected will influence the average reported rates of predation. The potential for bias is obvious: if you wanted to show high rates, you could use more high-predation villages. If you wanted to show low rates, you could use more low-predation villages. This might not affect your analysis of villagers’ attitudes, but it could certainly affect your measures of average effect. If my interpretation is correct, you will have to calculate your mean loss rates separately for the pre-selected high and low predation villages and give the number of such villages used as well as an estimate of the proportion of high- and low-predation villages in the area. There might also be some attention in the Discussion to why some villages have more predation than others and perhaps even a statistical comparison if there are sufficient data. If my interpretation is not correct, you need to be clearer about how you selected the villages. Please contact me if you need to discuss further this issue.

In addition, there are several grammatical problems that I have highlighted in an annotated version of the manuscript, which I have attached to my review.

I am therefore returning the manuscript again for minor revisions.

Reviewer 3 ·

Basic reporting

The editor asked me to comment and evaluate the statistical methods used in this manuscript, more particularly on the Information theoretic approach and AICc.
Overall, the statistical methods are valid and used properly. I only have some minor comments to clarify the Data Analysis section.

1) Line 137-141: More details should be provided concerning the reason why authors used model averaging.The authors should mention somewhere that the model with the lowest AICc is the best model, independently from being 2deltaAICc from the next best model. Then, it should be state clearly that if some models are within 2deltaAICc, it means that these models also have support to explain the data and should also be accounted for in the results/analysis. It is the reason why the authors used model averaging, i.e. to account for the support of more than 1 model. This will help neophytes to understand the use of Information theoretic approach (i.e. AIC).

2) Line 137-138: The deltaAICc (difference between 2 models) does not give you a measure of how much likely a model is the best one. The ratio between the model-normalized Akaike weight (or relative importance) of two MODELS can be interpreted as a likelihood. The same method can be used for the variable-normalized Akaike weight.

3) Line 141: I use Information theoretic approach and AICc regularly and never heard about the natural average method. The authors should provide a reference or add more details about this technique.

4) Line 142: This is a very technical question but why the authors used the 90%CI instead of the 95%CI? The 95%CI is more widely used.

5) Line 145: Could you also provide a reference to support your statement about the association of a variable to the response variable when the w+ is larger than 0.7? A variable can have a very high w+ but very low explanatory power (i.e. not within the 95 or 90%CI). The w+ of each variable helps to compare variables among each other by examining the ratio of w+.

6) Table 1: The last two variables should be inverted in order to be consistent (decreasing value of w+). Education_secondary and above should come before Education_elementary school.

Experimental design

No Comments

Validity of the findings

See basic reporting.

Additional comments

No Comments

---

## Round 0.3 · Minor Revisions

There are a few minor grammatical, spelling and clarity problems remaining in your manuscript. I have attached a copy with the problems indicated in yellow highlight and an inserted comma to suggest alternative wording and, in some cases, explain the problem. It should not take you long to make the changes and then the manuscript can be accepted.

---

## Round 0.4 · accepted · Accept

After reviewing your revised manuscript, I now consider it suitable for publication.